# Movement behavior remains stable in stroke survivors within the first two months after returning home

Roderick Wondergem[1,2,3]*, Martijn F. Pisters[1,2,3], Martijn W. Heijmans[4], Eveline J. M. Wouters[3,5], Rob A. de Bie[6], Cindy Veenhof[1,2,7], Johanna M. A. Visser-Meily[2,8]

**1** Center for Physical Therapy Research and Innovation in Primary Care, Julius Health Care Centers, Utrecht, The Netherlands, **2** Department of Rehabilitation, Physical Therapy Science and Sport, Brain Center, University Medical Center Utrecht, University Utrecht, Utrecht, The Netherlands, **3** Department of Health Innovations and Technology, Fontys University of Applied Sciences, Eindhoven, The Netherlands, **4** Department of Epidemiology and Biostatistics, Amsterdam UMC, Amsterdam Public Health Research Institute, VU University Medical Center, Amsterdam, The Netherlands, **5** Department of Tranzo, School of Social and Behavioral Sciences, Tilburg University, Tilburg, The Netherlands, **6** Department of Epidemiology and Caphri Research School, Maastricht University, Maastricht, The Netherlands, **7** Expertise Center Healthy Urban Living, Research Group Innovation of Human Movement Care, University of Applied Sciences Utrecht, Utrecht, The Netherlands, **8** Center of Excellence for Rehabilitation Medicine, Brain Center, University Medical Center Utrecht and De Hoogstraat Rehabilitation, Utrecht, The Netherlands

* r.wondergem@fontys.nl

## Abstract

### Background and purpose

The aim of this study is to investigate changes in movement behaviors, sedentary behavior and physical activity, and to identify potential movement behavior trajectory subgroups within the first two months after discharge from the hospital to the home setting in first-time stroke patients.

### Methods

A total of 140 participants were included. Within three weeks after discharge, participants received an accelerometer, which they wore continuously for five weeks to objectively measure movement behavior outcomes. The movement behavior outcomes of interest were the mean time spent in sedentary behavior (SB), light physical activity (LPA) and moderate to vigorous physical activity (MVPA); the mean time spent in MVPA bouts $\geq$ 10 minutes; and the weighted median sedentary bout. Generalized estimation equation analyses were performed to investigate overall changes in movement behavior outcomes. Latent class growth analyses were performed to identify patient subgroups of movement behavior outcome trajectories.

### Results

In the first week, the participants spent an average, of 9.22 hours (67.03%) per day in SB, 3.87 hours (27.95%) per day in LPA and 0.70 hours (5.02%) per day in MVPA. Within the entire sample, a small but significant decrease in SB and increase in LPA were found in the

**Data Availability Statement:** Data cannot be shared publicly because of data contain potentially identifying or sensitive patient information. Data are available from the UMC Utrecht Ethics

Committee info@umcutrecht.nl for researchers who meet the criteria for access to confidential data.

**Funding:** This study was funded by the Netherlands Organization for Scientific Research (NWO), Doctoral grant for Teachers, 023.003.136 to RW. The funder had no role in study design, data collection and analysis, decision to publish, or preparation of the manuscript.

**Competing interests:** The authors have declared that no competing interests exist.

**Abbreviations:** ADL, activities of daily living; LPA, light physical activity; METs, metabolic equivalents; MVPA, moderate to vigorous physical activity; SB, sedentary behavior.

first weeks in the home setting. For each movement behavior outcome variable, two or three distinctive subgroup trajectories were found. Although subgroup trajectories for each movement behavior outcome were identified, no relevant changes over time were found.

## Conclusion

Overall, the majority of stroke survivors are highly sedentary and a substantial part is inactive in the period immediately after discharge from hospital care. Movement behavior outcomes remain fairly stable during this period, although distinctive subgroup trajectories were found for each movement behavior outcome. Future research should investigate whether movement behavior outcomes cluster in patterns.

## Introduction

The majority of stroke survivors are discharged to the home setting immediately after hospital care [1]. Following a stroke, cardiovascular event rates are high. Premature death and disability rates are higher after a recurrent event than after the first stroke [2,3]. Secondary lifestyle interventions are important and have been demonstrated to be effective in reducing systolic blood pressure, one of the strongest risk factors for both first and recurrent stroke [4,5]. An important lifestyle intervention that can favorably influence cardiovascular risk is changing movement behaviors [6]. Movement behaviors during waking hours include sedentary behavior (SB), and physical activity (PA) [7]. Within PA, the intensities of light physical activity (LPA) and moderate to vigorous physical activity (MVPA) can be distinguished. SB is defined as "any waking activity characterized by an energy expenditure of $\leq$ 1.5 metabolic equivalents (METs) and a sitting or reclining posture" [8], LPA consists of activities between 1.5 and 3.0 METs, and MVPA consists of all activities > 3.0 METs. In general, stroke survivors are highly sedentary and inactive compared to healthy peers [9].

Various movement behavior outcomes have shown associations with health risk and functional decline [10,11]. The composition of movement behavior during waking hours (the relative amounts of SB, LPA and MVPA during waking hours), the continuity of SB (interrupted or prolonged SB), and the continuity of MVPA (bouts $\geq$10 minutes) are important modifiable risk factors to improve cardiovascular health. High amounts of SB and low amounts of MVPA are independent risk factors for cardiovascular disease incidence and premature mortality. MVPA should occur in bouts of at least ten minutes to contribute to the recommended 150 minutes per week spent in MVPA [12]. Additionally, long uninterrupted sedentary bouts are related to cardiovascular risks [13]. Interrupting SB after 20 minutes has been found to have a positive influence on glucose levels in overweight people [14], and interruption after 30 minutes decreased the systolic blood pressure of stroke survivors [15], thus providing cardiovascular health benefits.

Few longitudinal studies have investigated changes in movement behavior during waking hours in stroke survivors. Two small longitudinal studies focusing on the first three months after discharge from a rehabilitation hospital stroke unit found significant increases in both LPA and MVPA [16]. In contrast, another study found an increase in SB [17]. To date, all studies investigating the course of movement behavior outcomes up to the first year after stroke have used averaged group data and found no changes over time [9,16,18,19]. However, recovery after stroke is not a one-size-fits-all principle; it is characterized by individual patterns [20]. Previous studies have demonstrated variation in the trajectories of physical and

psychosocial health-related quality of life [21] and functional recovery [22] within the first year after stroke. In healthy populations, SB and MVPA were found to have four to seven subgroup trajectories each [23,24]. Because the stroke recovery process is heterogeneous, different subgroup trajectories of movement behavior outcomes can be expected.

The hypothesis is that a decrease in total sedentary time, increased interruption of SB, and increases in LPA and MVPA will occur in the initial period after discharge. These outcomes might be expected because most functional recovery occurs within the first few weeks after stroke [20], and most stroke survivors still receive professional support or rehabilitation during that period [25]. Moreover, during those initial weeks, health care professionals provide information regarding modifiable risk factors, including movement behaviors [26]. Additionally, it is expected during the period shortly after this life event, are especially motivated to improve their lifestyle to prevent recurrent events [27]. Therefore, trajectories of changes in movement behavior outcomes are expected. However, knowledge is currently lacking regarding the course of movement behavior outcomes shortly period after discharge.

Stroke recovery is heterogeneous, and average group data, assumes a one-size-fits-all principle, possible changes in movement behavior outcomes in subgroup trajectories maybe overlooked. Therefore, subgroup trajectories of change in movement behavior outcomes need to be investigated, since they are expected. To identify potential subgroup trajectories, data-driven analyses are needed. Latent class growth analysis is a method whereby participants are assumed to belong to a single class but which class is not known [28]. This approach will extend our understanding of subgroup trajectories of change in movement behavior outcomes. Once these subgroup trajectories are known, associations will need to be explored. Currently, only a few associations are known with regard to movement behavior outcomes. Lower walking speed and walking capacity, balance problems, presence of depression and poorer quality of life associated with accelerometer activity counts [9]. Additionally, higher age, being a man, higher cardiorespiratory fitness, lower levels of fatigue, higher level of self-efficacy, presence of depression and higher health related quality of life were factors associate with higher levels of PA [29]. Lower walking speed was found to associated with higher amount of sedentary time and long prolonged bouts [30], less functional independence with high amounts of sedentary behavior and prolonged bouts, stroke severity with high amounts of sedentary behavior and age with more prolonged sedentary behavior [19]. Although these studies provide preliminary information deeper understanding of factors related with single movement behavior outcomes is needed.

Therefore, the aim of the current study is 1) to investigate changes in both the distribution (SB, LPA, and MVPA) and accumulation (bouts) of movement behavior during waking hours for the entire sample, and 2) to detect possible subgroup trajectories within each movement behavior outcome within the first two months after discharge from hospital care to the home setting in first-time stroke patients. Once these subgroup trajectories are known, 3) associated patient characteristics will be explored.

## Method

### Design and participants

Data cannot be shared publicly because data contain potentially identifying or sensitive patient information. Data are available from the UMC Utrecht Ethics Committee info@umcutrecht.nl for researchers who meet the criteria for access to confidential data. Eligible participants were recruited between February 2015 and April 2017 from four participating stroke units in the Netherlands. This prospective longitudinal cohort study, carried out after discharge from a hospital directly to the patients' own home settings, specifically recruited persons who had

suffered a clinically confirmed first-ever stroke and who had been independent in ADL before stroke (Barthel index score >18 [31]). Other inclusion criteria were age over eighteen years, ability to sustain a conversation (Utrecht Communication Assessment score > 4 [32]) and at least the ability to walk with supervision after stroke (Functional Ambulation Categories score >2 [33]). People with a subarachnoid hemorrhage were excluded. The written informed consent of the participants was obtained at the stroke unit. All participants gave written informed consent. The study was approved by the Medical Ethics Research Committee of the University Medical Center Utrecht (study number 14/76).

## Measurements and procedures

After discharge from the hospital, participants were visited at home within three weeks after discharge. During this visit, walking speed, balance, and levels of activity and participation were obtained. Participants received an accelerometer to objectively measure movement behavior during waking hours. The participants wore the accelerometer for five consecutive weeks before returning the device by mail.

## Independent characteristics

The personal characteristics obtained were the age and sex of the participants and whether they lived alone. Stroke severity was measured with the National Institute of Health Stroke Scale (NIHSS) (range 0–42, with higher scores indicating more severe stroke symptoms). The NIHSS discerns three subgroups: 1) no stroke symptoms (0 points); 2) minor stroke (1–4 points);and 3) moderate to severe stroke ($\geq$ 5 points) [34]. Stroke services are a form of integrated care that has been established during the last decade. The aim of stroke services is to improve health outcomes and processes of care by connecting the acute, rehabilitative, and chronic phases of stroke care [35,36]. In a typical Dutch stroke service, the hospital, rehabilitation center (in- or outpatient care), and primary physiotherapy care are represented. Information about physiotherapy care was obtained from medical records and verified by asking the participant. Three options were possible no physiotherapy care, primary physiotherapy care and outpatient multidisciplinary rehabilitation care that included physiotherapy. The Montreal Cognitive Assessment was used to assess cognitive functioning [37,38]. Scores were dichotomized into normal ($\geq$ 26) or impaired (< 26) cognitive function. The Hospital Anxiety and Depression Scale was used to assess the presence of depression and anxiety symptoms. Each subscore was dichotomized into the presence ($\geq$ 8 points) or absence (< 8 points) of depression or anxiety symptoms [39,40]. The Late-Life Function and Disability Instrument Computerized Adaptive Test activity limitations and participation restriction subscales scores were obtained [41]. Physical functioning was measured with the physical functioning subdomain of the Stroke Impact Scale (SIS) 3.0 [42,43] The physical functioning subdomain consists ten questions regarding ADL, eight regarding mobility, and five regarding hand function [42,43]. As recommended, scores were calculated as percentages of the total points, resulting in a range from 0 to 100. Lower scores indicate lower levels of physical functioning. Balance was tested with the Berg Balance Scale [44]. Walking speed was obtained using the five-meter walk test [45]. All outcome measures are valid and reliable in a stroke population.

## Accelerometer

Movement behavior during waking hours was objectively measured with an Activ8 accelerometer. The Activ8 is a triaxial accelerometer (30x32x10 mm, 20 grams). Participants were instructed to carry the accelerometer in the front pocket of their pants on the unaffected leg the whole day during waking hours. Only when taking a shower or swimming were

participants allowed to remove the device. Clear wearing instructions were given, and participants were asked to record in an activity log the time when they put on the Activ8 in the morning and the time when they removed it. The device can detect SB (combination of reclining and sitting), standing, walking, cycling and running and provide corresponding MET values. The Activ8 measures with a sampling frequency of 12.5 Hz, an epoch of 1 second and a sample interval of 5 seconds. Every 5 minutes, a summary was stored of the different postures and their respective MET values. The device is able to store data for sixty days, and its battery life is at least 30 days [46]. The Activ8 has been validated in a community-living stroke population in terms of postures and in a healthy population in terms of energy expenditures [47,48].

## Movement behavior outcomes

Individual days were screened, and nonwear time was removed from the data files using starting and stopping times. Using SPSS, the most important and recommended movement behavior outcomes were calculated [49]. The mean times spent in SB, LPA, and MVPA in hours per day were computed by summation and divided by the number of wearing days per individual [7]. The mean time of MVPA accumulated in bouts ≥ 10 minutes was calculated. An MVPA bout was defined as 10 or more consecutive minutes of MVPA, with allowance for interruptions of no more than 2 minutes [50]. For each individual, the weighted median sedentary bout length was calculated [49]. The weighted median sedentary bout is the sedentary bout that corresponds to 50% of the total sedentary time [49]. The weighted median sedentary bout length is more sensitive to change than the total time spent in SB [51]. Bouts are ordered from smallest to largest, and for example, if an individual has spent eight hours sedentary, this measure represents the length of the bout that contains the four hour timepoint. If this would be 20 minutes, it means that individuals spend 50% their SB time in bouts ≥ 20 minutes. The lower the weighted median sedentary bout is, the more frequently interrupted the SB.

## Data analysis

SPSS version 25.0 [52] was used for descriptive statistics, which are expressed as the means and standard deviations. The multivariate imputation by chained equation procedure was used to impute (multiple) missing values [53]. In our data set, missing data were not dependent on descriptive characteristics; therefore, data were assumed to be missing completely at random, and multivariate imputation by chained equations was therefore indicated to increase statistical power [54]. Multivariate imputation by chained equations was performed by models to predict missing values for a given variable based on all other observed variables. Five imputed data sets were created and combined to create a pooled set using Rubin's rules [55].

To investigate the average group movement behavior change within the first weeks after discharge to the home setting, generalized estimation equation were employed [56]. Latent class growth analysis was performed with Mplus version 8.1 [57] to identify clinically relevant homogeneous subgroups of stroke survivors that followed different trajectories of movement behavior. For each movement behavior outcome, latent class growth analysis was performed. Latent class growth analysis uses latent variables to estimate differences in mean changes over time in different subgroups, taking into account individual longitudinal trajectories. The trajectories within the subgroups were kept homogenous. The fit of the models was tested by comparing models with two, three, four and five subgroups. Both linear and quadric trajectories were modeled and compared. Statistical considerations for finding the most appropriate model included a Bayesian information criterion, entropy values and the bootstrap likelihood ratio test [58–60]. The lower the Bayesian information criterion score, the better the fit of the model [60]. When bootstrap likelihood ratio test was significant (p<0.05), the model with k-

subgroups had a better fit than the model with k-1 trajectory subgroups [28,60]. The entropy statistic was used for the reliability of the subgroup trajectories. Entropy scores above 0.8 are preferred [28]. When less than 5% of the sample was assigned to a subgroup trajectory, a k-1 subgroup trajectory model was chosen [61].

If more than two subgroup trajectories were found based on the latent class growth analysis, trajectories were merged into two clinically relevant subgroups. To determine the characteristics associated with a single subgroup trajectory, logistic regression analyses were performed. Odds ratios were calculated to identify candidate factors using univariate analyses. The related variables were tested for multicollinearity (Pearson's r < 0.70) and effect modification (variance inflation factor >4) [62]. Significantly associated characteristics (p<0.1) were entered into a multiple backward logistic regression analysis.

## Results

In total, 180 people with stroke agreed participation when discharged from the hospital to the home-setting. With twenty persons it was not possible to make an appointment within three weeks, fifteen refused further participation, three were unable to contact, one was to ill and one died before our visit. Resulting in140 participants included in this study. The mean age of the population was 66.4 years, and the majority of the population was male (66.4%). Stroke severity two days after stroke was mild in 63.6% of the population. Other characteristics are presented in Table 1.

In total, 4.81% of the movement behavior outcomes were missing and imputed. The mean Activ8 wear time in week one was 13.78 hours per day and did not change during the subsequent four assessment weeks. The overall mean sedentary time during the five consecutive weeks was high, with a mean of 9.22 hours in week one, with a significant average decrease of 0.06 hours per week, leading to 8.9 hours in week five. The time spent in LPA was 3.87 in week one, increasing significantly by 0.05 hours per week, leading to 4.08 hours. All other movement behavior outcomes remained stable over time. The mean time spent in MVPA was 0.70 hours in week one, and MVPA accumulated in bouts ≥ 10 minutes in week one was 0.29 hours. A mean weighted median sedentary bout of 21.82 minutes was found in week one. All movement behavior outcomes by week and all generalized estimating equations outcomes can be found in Table 2.

Different amount of subgroup trajectories were found for movement behavior outcome. (see Table A in S1 File, for the Bayesian information criterion, entropy and bootstrap likelihood ratio test outcomes for each subgroup trajectory). Although the fit of most models favored a four or five subgroup model, some subgroup trajectories contained too few individuals to be considered clinically relevant (less than 5% of the total sample). Consequently, two subgroup trajectories were determined for SB and LPA. Three subgroups were found for MVPA, MPVA spent in bouts ≥ 10 minutes, and weighted median sedentary bouts. For total SB, LPA and MVPA, quadratic trajectories are presented because lower Bayesian information criterion values and higher entropy values were found. Linear trajectories were presented for weighted median sedentary bouts and MVPA accumulated in bouts ≥ 10 minutes. The Bayesian information criterion, entropy, bootstrap likelihood ratio test, intercepts and slopes are presented in Table 3. All presented subgroup trajectories had entropy scores above 0.8.

The stroke survivors allocated to the two subgroup SB trajectories spent a mean of 7.92 and 9.94 hours in SB, respectively. In this manner, 64.3% were classified as 'highly sedentary' and 35.7% as 'less sedentary'. The time spent in LPA varied between 3.17 and 5.02 hours. A total of 65.7% of the participants were classified as 'nonmovers', and 34.3% were classified as 'movers'. Three subgroups were found regarding MVPA and MVPA spent in bouts ≥10 minutes. Only

**Table 1. Participant characteristics expressed as mean±SD, median [IQR], or n (%).**

| Characteristics (N = 140) | % or mean±SD |
|---|---|
| **Personal characteristics** | |
| Males | 66.4 |
| Age (years) | 67.1±10.8 |
| Living alone | 18.6 |
| **Stroke characteristics** | |
| Time since stroke (days) | 19.6±5.6 |
| Infarction | 91.4 |
| Side of stroke | |
| Left | 52.9 |
| Right | 42.1 |
| Both | 2.1 |
| Unknown | 2.9 |
| Stroke severity day 2 after stroke | |
| No symptoms (NIHSS 0) | 15.0 |
| Minor stroke symptoms (NIHSS 1 to 4) | 63.6 |
| Moderate to severe stroke symptoms (NIHSS $\geq$5) | 21.4 |
| **Psychological characteristics** | |
| Cognitive functioning [a] | |
| Impaired cognitive function (MOCA $\leq$25)[a] | 39.1 |
| Depression (HADS-D) | 13.7 |
| Anxiety (HADS-A) | 16.7 |
| **Physiotherapy care** | |
| Outpatient multidisciplinary rehabilitation, including physiotherapy [a] | 12.8 |
| Primary care physiotherapy [a] | 33.6 |
| No physiotherapy [a] | 53.6 |
| **Functional ability** | |
| Walking speed (m/s)[a] | 1.03±0.24 |
| Limited community walker ($\geq$0.93 m/s)[a] | 31.4 |
| LFDI-CAT activity limitations [a] | 58.9±10.8 |
| Physical functioning (SIS) [a] | 93.8 [82.3–98.9] |
| LLFDI-CAT participation restrictions [a] | 48.9±10.7 |
| Balance (BBS) | 55 [52.2–56] |

% = percentage, SD = standard deviation, IQR = interquartile range, NIHSS = National Institutes of Health Stroke Scale, MOCA = Montreal Cognitive Assessment, HADS = Hospital Anxiety and Depression Scale, m/s = meters per second, LLFDI-CAT = Late-Life Function and Disability Instrument Computer Adaptive Testing, SIS = Stroke Impact Scale, BBS = Berg Balance Scale

[a] Assessments were carried out in the participant's home setting within three weeks after discharge from inpatient care (hospital or inpatient rehabilitation).

Higher scores indicate better outcomes except for walking speed.

10.7% were identified as 'highly active', while 34.3% were 'active', and 55% were 'inactive'. The results for time spent in MVPA bouts $\geq$10 minutes was slightly worse. Altogether, 10% of the participants could be classified as 'prolongers', 52.8% as 'intermediate' and 37.1 as 'interrupters', with weighted median sedentary bout lengths of 50 minutes, 24 minutes and 11 minutes, respectively. All outcomes can be found in Table 3.

**Table 2. Movement behavior outcomes in the overall group and generalized estimating equations analyses adjusted for wear time.**

| Movement behavior outcome | Week 1 [95%CI] | Week 2 [95%CI] | Week 3 [95%CI] | Week 4 [95%CI] | Week 5 [95%CI] | B [95%] |
|---|---|---|---|---|---|---|
| **Sedentary (hours/day)** | 9.22 [8.94–9.46] | 9.18 [8.87–9.49] | 9.25 [8.96–9.54] | 9.00 [8.71–9.30] | 8.99 [8.73–9.26] | -0.06 [-0.11–0.02]* |
| **LPA (hours/day)** | 3.87 [3.60–4.13] | 3.98 [3.71–4.25] | 3.93 [3.68–4.18] | 4.06 [3.79–4.33] | 4.08 [3.83–4.33] | 0.05 [0.01–0.09]* |
| **MVPA (hours/day)** | 0.70 [0.61–0.78] | 0.69 [0.60–0.78] | 0.71 [0.62–0.80] | 0.65 [0.57–0.74] | 0.73 [0.65–0.82] | 0.01 [-0.02–0.02] |
| **MVPA accumulated in bouts≥10 minutes (hours/day)** | 0.29 [0.22–0.35] | 0.25 [0.20–0.30] | 0.25 [0.20–0.30] | 0.27 [0.22–0.32] | 0.28 [0.23–0.33] | 0.00 [-0.01–0.01] |
| **Weighted median sedentary bout (minutes)** | 21.82 [19.71–23.93] | 20.71 [18.49–22.92] | 21.85 [19.64–24.06] | 21.53 [19.46–23.60] | 21.16 [19.17–23.16] | -0.05 [-0.44–0.34] |
| **Wear time (hours/day)** | 13.78 [13.70–13.87] | 13.85 [13.76–13.94] | 13.89 [13.79–13.99] | 13.81 [13.72–13.91] | 13.82 [13.72–13.92] | n.a. |

CI = confidence interval, LPA = light physical activity, MVPA = moderate to vigorous physical activity

*P<0.05.

Figs 1–5 show subgroup trajectories of all movement behavior outcomes. A small but significant decrease in sedentary time was found in the subgroup trajectory of highly sedentary people. The inactive subgroup increased slightly in time spent in MVPA, whereas the active subgroup slightly decreased. All other subgroup trajectories of movement behavior outcomes remained stable within the first two months.

The 'active' and 'highly active' subgroup trajectories for both MVPA and MVPA spent in bouts ≥10 minutes were merged together since the participants in both subgroups were sufficiently active, since international guidelines recommend at least 150 minutes per week of accumulated moderate to vigorous physical activity (MVPA) [12]. Additionally, 'intermediate' and 'interrupters' subgroup trajectories for the weighted median sedentary bout length were merged. Although there are no clear cut-off values available for the interruption of SB, interruption after thirty minutes was been found to have health benefits in people with stroke

**Table 3. Model fit indices for the selected subgroup trajectories for each movement behavior outcome, n = 140.**

| | Subgroups (n) | Intercept for subgroup | Linear slope | Quadric slope | BIC | Entropy | BLRT |
|---|---|---|---|---|---|---|---|
| **Sedentary behavior (hours/day)** | Highly sedentary = 90 | 9.94 | 0.25 | -0.06* | 2343.72 | 0.87 | <0.01 |
| | Less sedentary = 50 | 7.92 | -0.37 | 0.07 | | | |
| **LPA (hours/day)** | Non-movers = 92 | 3.17 | -0.10 | 0.03 | 2192.88 | 0.82 | <0.01 |
| | Movers = 48 | 5.02 | 0.33 | -0.06 | | | |
| **MVPA (hours/day)** | Inactive = 77 | 0.43 | -0.08* | 0.01* | 2192.88 | 0.82 | <0.01 |
| | Active = 48 | 1.02 | -0.08 | 0.01 | | | |
| | Highly active = 15 | 1.43 | 0.21 | -0.04* | | | |
| **MVPA bouts≥10 min (hours/day)** | Inactive = 89 | 0.10 | 0.01 | n.a. | -83.89 | 0.93 | <0.01 |
| | Active = 42 | 0.40 | 0.01 | n.a. | | | |
| | Highly active = 9 | 1.05 | -0.01 | n.a. | | | |
| **Weighted median sedentary bout length (min)** | Prolongers = 14 | 49.97 | -1.64 | n.a. | 5151.92 | 0.91 | <0.01 |
| | Intermediate = 74 | 23.90 | 0.17 | n.a. | | | |
| | Interrupters = 52 | 11.00 | 0.08 | n.a. | | | |

BIC = Bayesian information criteria, BLRT = bootstrap likelihood ratio test, LPA = light physical activity, MVPA = moderate to vigorous physical activity,

min = minutes

*p<0.05.

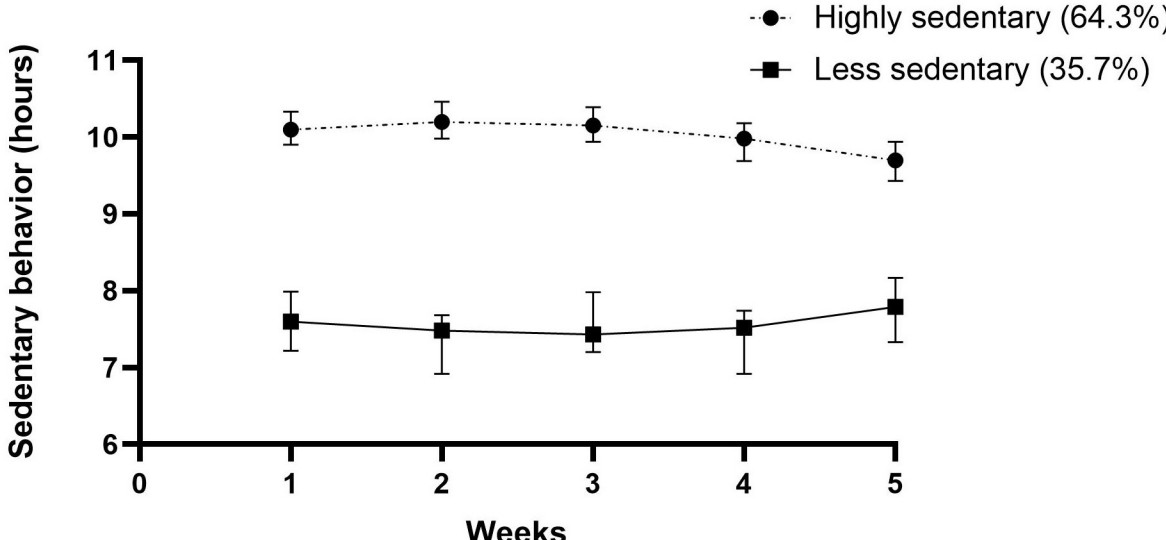

**Fig 1. Sedentary time in hours per day between three weeks and eight weeks after discharge from the hospital in stroke survivors.**

[63,64]. The distribution of individuals to the different subgroups is presented in Table B in S1 File. The results show that the different movement behavior outcomes reveal distinct trajectories. For example, 53.6% of the population was highly sedentary and classified as nonmovers, and 35.7% was inactive and highly sedentary.

The results of the univariate analyses per movement behavior subgroup are presented in Table 4. The results of the multiple logistic regression analyses are presented in Table 5. No

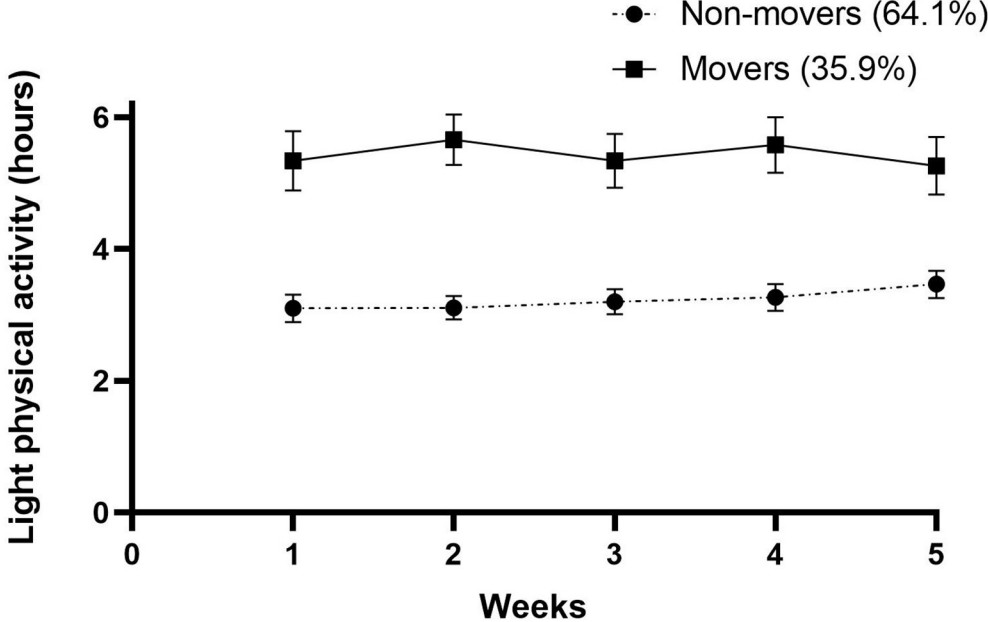

**Fig 2. Light physical activity in hours per day between three weeks and eight weeks after discharge from the hospital in stroke survivors.**

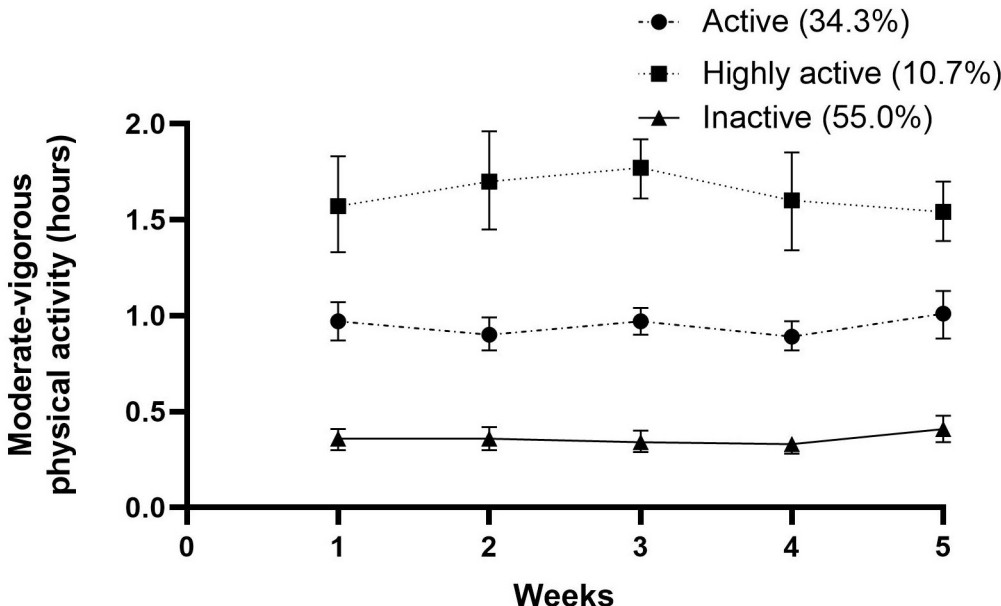

**Fig 3. Moderate to vigorous physical activity in hours per day between three weeks and eight weeks after discharge from the hospital in stroke survivors.**

associations were found regarding SB. Factors associated with nonmovers were living with another person and impaired cognitive function. Being male, and younger and having fewer activity limitations were associated with both active groups (MVPA and MVPA spent in bouts ≥10 minutes). Living alone and being a community walker were only associated with the active MVPA group. Factors associated with prolongers were more severe stroke symptoms, cognitive impairment and not being a community walker.

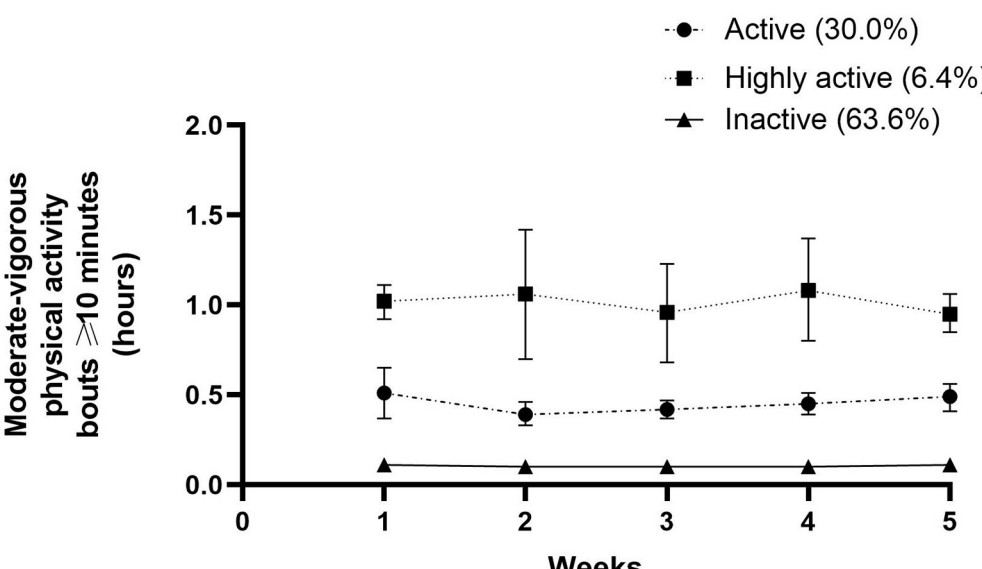

**Fig 4. Moderate to vigorous physical activity bouts (≥10 minutes) in hours per day between three weeks and eight weeks after discharge from the hospital in stroke survivors.**

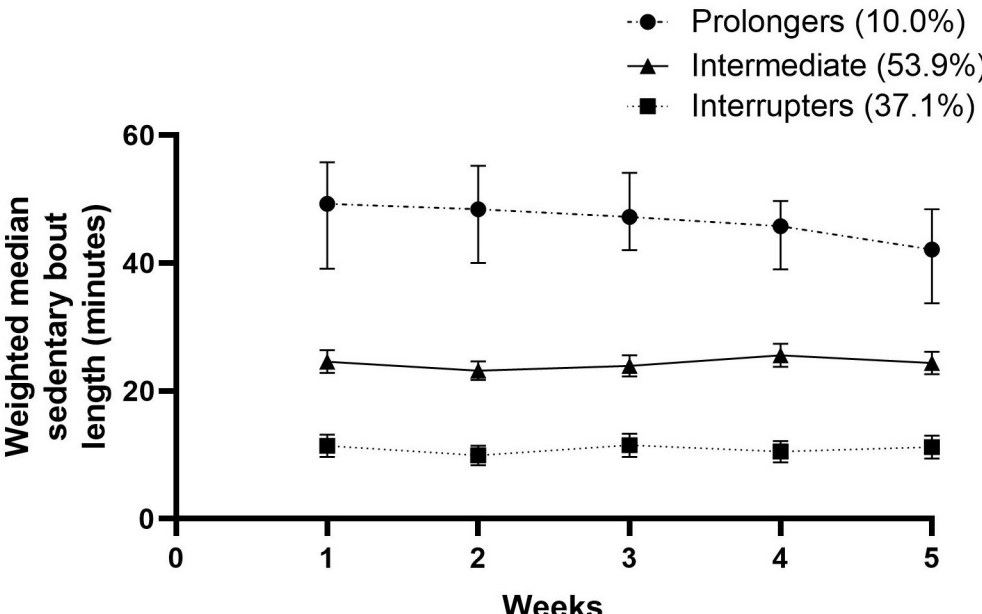

**Fig 5. Weighted median sedentary bout in minutes per day between three weeks and eight weeks after discharge from the hospital in stroke survivors.**

## Discussion

This study investigated changes in movement behavior outcomes and possible subgroup trajectories using objective and continuous measurement in 140 participants within the first two months after discharge from the hospital to the home setting after a first stroke. Overall, SB decreased very slightly, and LPA showed a small increase in time. Distinct subgroup trajectories were found for all movement behavior outcomes. Small changes within subgroup trajectories for SB and MVPA were found. For all other movement behavior outcomes, the identified subgroup trajectories remained stable. Individuals were distributed into different subgroups according to movement behavior outcomes. Characteristics associated with the different subgroups were explored. No associated characteristics were found regarding SB.

On average, our sample showed SB results comparable to a Dutch older adult population [65]. In our sample, that the majority of the people were highly sedentary, exceeding 9.5 hours. The relationship between sedentary time and mortality was more pronounced when sedentary periods exceeded 9.5 hours [66]. Therefore, the reduction of SB should be a secondary prevention target for people with stroke. On average, our sample engaged in 42 minutes of MVPA per day, which is high. It is known that the Dutch population is more active than its European peers[67]. In other stroke survivors, the same amount of MVPA was found (44 minutes) [18]. Although, the average amount of MVPA was high, a substantial part of the population was found to be inactive. Particularly in terms of MVPA accumulated in bouts ≥10 minutes.

Although a significant decrease in SB and an increase in time spent in LPA were found within the first two months after discharge, the changes were small. However, it was recently found that higher levels of physical activity, including light physical activity, and less time spent in SB reduce the risk of premature death in a dose-response manner [66]. Therefore, even this small change in LPA and SB are considered relevant. Nevertheless, the absolute amount of SB was still high. A previous study (N = 10) found an increase of forty minutes in absolute activity during the day within the first six weeks after discharge to the home setting

**Table 4. Associated factors with highly sedentary, non-movers, active and prolongers using univariate analyses.**

| | HIGHLY SEDENTARY | NON-MOVERS (LPA) | ACTIVE (MVPA) | ACTIVE (MVPA BOUTS> 10 MIN) | PROLONGERS (WMSB) |
|---|---|---|---|---|---|
| INDEPENDENT VARIABLES | OR (95%CI) | OR (95%CI) | OR (95%CI) | OR (95%CI) | OR (95%CI) |
| **PERSONAL CHARACTERISTICS** | | | | | |
| MALE | 1.29 (0.61–2.71) | 1.30 (0.63–2.71) | 2.63 (1.25–5.54)*** | 5.13 (2.09–12.92)*** | 0.90 (0.28–2.86) |
| LOWER AGE (YEARS) | 1.00 (0.97–1.04) | 1.02 (0.98–1.05) | 1.06 (1.02–1.10)*** | 1.04 (1.01–1.08)** | 0.95 (0.9–1.01)* |
| LIVING TOGETHER | 1.71 (0.73–4.07) | 0.44 (0.19–1.05)* | 2.28 (0.95–5.46)* | 0.91 (0.37–2.22) | 0.71 (0.15–3.38) |
| **STROKE / CARE CHARACTERISTICS** | | | | | |
| MORE SEVERE STROKE SYMPTOMS (NIHSS) | 1.12 (1.00-.125) | 0.90 (0.81–1.01)* | 1.02 (0.93–1.12) | 1.02 (0.93–1.11) | 0.90 (0.81–0.99)** |
| INFARCTION | 1.12 (0.32–3.03) | 0.96 (0.27–3.35) | 1.71 (0.49–5.97) | 1.16 (0.33–4.06) | 0.52 (0.10–2.64) |
| NO PT CARE | 1.37 (0.68–2.74).68–2.74 | 0.95 (0.47–1.92) | 0.53 (0.27–1.05)* | 0.69 (0.34–1.39) | 1.25 (0.41–3.77) |
| **PSYCHOLOGICAL AND COGNITIVE FACTORS** | | | | | |
| COGNITIVE IMPAIRED (MOCA ≤25) | 0.83 (0.41–1.67) | 2.09 (1.01–4.33)** | 0.97 (0.49–1.89) | 1.38 (0.69–2.77) | 4.68 (1.23–17.84)** |
| ABSENCE DEPRESSION (<8 HADS-D) | 1.68 (0.57–4.98) | 0.46 (0.14–1.48) | 1.96 (0.70–5.5) | 1.30 (0.46–3.66) | 0.16 (0.05–0.52)*** |
| ABSENCE ANXIETY (<8 HADS-A) | 0.56 (0.23–1.38) | 1.00 (0.39–2.56) | 1.07 (0.44–2.65) | 1.08 (0.42–2.76) | 0.64 (0.16–2.51) |
| **PHYSICAL FUNCTIONING** | | | | | |
| NONCOMMUNITY WALKER (≥0.93 m/s) | 1.04 (0.05–2.19) | 1.18 (0.55–2.51) | 0.11(0.04–0.28)*** | 0.14 (0.04–0.38)*** | 3.33 (1.08–10.29)** |
| LOWER ACTIVITY LIMITATIONS (SIS) | 0.99 (0.67–1.02) | 1.00 (0.97–1.03) | 0.92 (0.89–0.96)*** | 0.91(0.87–0.96)*** | 1.03 (0.98–1.06) |
| LOWER FUNCTIONING OF BALANCE (BBS) | 0.96 (0.88–1.05) | 1.01 (0.94–1.09) | 0.77 (0.66–0.90)*** | 0.64 (0.050–0.81)*** | 1.03 (0.95–1.13) |

CI = confidence interval, LPA = Light physical activity, MVPA = Moderate to vigorous physical activity, WMSB = Weighted median sedentary bout length, OR = odds ratio, CI = confidence interval, PT = physiotherapy. Age, less severe stroke symptoms, lower activity limitations and balance were analyzed as continues variables. NIHSS = National Institutes of Health Stroke Scale, MOCA = Montreal Cognitive Assessment, HADS-D = Hospital Anxiety and Depression Scale depression subscale, HADS-A = Hospital Anxiety and Depression Scale anxiety subscale, m/s = meters per second, SIS = Stroke Impact Scale, BBS = Berg Balance Scale.

*p<0.1

**p<0.05

***p<0.01

[16]. However, this improvement was compared to the absolute activity before discharge. When comparing activity at two weeks after discharge with activity at six weeks after discharge, an increase of only twenty minutes was found. The same increases were observed in another study regarding step count and time spent walking between one and three months after discharge [17]. We also found an LPA increase of twenty minutes. Therefore, it seems that after stroke, people increase their level of LPA in the short term. Regarding SB, conflicting results were found in the literature. In our sample, SB decreased while in another study with a small sample size sitting/reclining time increased [17]. However, in that study, sleep time was included in the sitting/reclining time. Therefore, it remains unknown whether SB, sleep time, or both increase within the first six months after discharge to the home setting [17].

The differences in the distribution and accumulation of movement behavior during the day are interesting. Over 60% of the sample was assigned to a subgroup trajectory with a mean sedentary time per day reaching almost ten hours out of fourteen hours wear time. This indicates high amounts of SB. Prolonged bouts are more difficult to interpret since there is not a given

**Table 5. Associated characteristics per movement behavior pattern using multiple logistic regression.**

| | SEDENTARY | NONMOVER (LPA) | ACTIVE (MVPA) | ACTIVE (MVPA BOUTS> 10MIN) | PROLONGER (WEIGHTED MEDIAN SEDENTARY BOUTS) |
|---|---|---|---|---|---|
| | OR (95%CI), P-value | OR (95%CI)5% | OR (95%CI) | OR (95%CI), P-value | OR (95%CI), P-value |
| **INDEPENDENT VARIABLES** | | | | | |
| **PERSONAL CHARACTERISTICS** | | | | | |
| MALE | | | 3.35 (1.39–8.08)** | 6.14 (2.37–15.92)** | |
| LOWER AGE (YEARS) | | | 1.05 (1.02–1.09)** | 1.05 (1.02–1.09)** | |
| LIVING ALONE | | 0.40 (0.22–0.74)** | 8.49 (2.22–32.44)** | | |
| **STROKE CHARACTERISTICS** | | | | | |
| LESS SEVERE STROKE SYMPTOMS (NIHSS) | | | | | 0.87 (0.77–0.99)* |
| **PSYCHOLOGICAL AND COGNITIVE FACTORS** | | | | | |
| COGNITIVE IMPAIRED (MOCA ≤25) | | 2.33(1.20–4.51)** | | | 5.02 (1.54–16.37)** |
| **PHYSICAL FUNCTIONING** | | | | | |
| NON-COMMUNITY WALKER (≥0.93 M/S) | | | 0.17 (0.06–0.55)** | | 3.11(1.15–8.44)* |
| LOWER ACTIVITY LIMITATIONS (SIS) | | | 0.94 (0.90–0.98)** | 0.96 (0.93–0.99)* | |

CI = confidence interval, LPA = Light physical activity, MVPA = Moderate to vigorous physical activity, WMSB = Weighted median sedentary bout length, OR = odds ratio, CI = confidence interval

Age, less severe stroke symptoms and lower activity limitations were analyzed as continues variables. NIHSS = National Institutes of Health Stroke Scale,

MOCA = Montreal Cognitive Assessment, m/s = meters per second, SIS = Stroke Impact Scale.

*p<0.05

**p<0.01

cut-off value available yet. However, the majority of the group had a weighted median bout of over 20 minutes, indicating that over50% of total sedentary time is spent in prolonged bouts. Interruption after 20 minutes of SB has been found to have a positive influence on glucose levels in overweight people [14].

Additionally, over 90% of the population did not reach sufficient amounts of MVPA accumulated in bouts of at least 10 minutes. Differences in the changes among the subgroup trajectories were found. Participants in the highly sedentary subgroup trajectory decreased their amount of sedentary time, and those in the inactive group increased their MVPA time. Both changes, in theory, can reduce the risk of premature death, although the changes are small [66].

Remarkably, we found no patient characteristics that were associated with highly sedentary behavior. A recent study, which pooled data from nine studies identifying associations with sedentary time after stroke, found that sedentary time could not be explained by demographic or stroke-related variables[30]. It identified only slower walking speed as a significant factor associated with higher amounts of SB. In our sample, people were discharged immediately to the home-setting and had a relatively high walking speed, whereas the study of Hendrickx et al included participants with greater diversity of walking speed. This could explain why walking speed was not identified as a factor associated with SB in our sample. Although living alone

was associated with the total MVPA time, it was not associated with MVPA accumulated in bouts ≥10 minutes. It seems that people who live alone spend time in MVPA during their ADLs and devote less leisure time MVPA in such forms as exercise or sports. More severe stroke symptoms, cognitive impairment and not being a community walker were associated with prolongers in our study. These outcomes are in line with previous studies of people with stroke, although those studies found associations with walking speed, more severe stroke symptoms and self-reported ADLs and sedentary bouts [30,68]. The association between cognitive impairment and nonmovers and prolongers is interesting since no associations were found with total sedentary time or MVPA in our sample. A study including older adults found that higher amounts of SB were associated with lower cognitive function when MVPA was not taken into account; however, no association was found after adjustment for MVPA. This indicates the importance of investigating movement behavior patterns and not just single movement behavior outcomes. Additionally, it could be that the health benefits of enough MVPA are counterbalanced by high amounts of SB.

Trajectories of single movement behavior outcomes overlap; however, they are largely unique. For example, 54% of the people who were highly sedentary were nonmovers but only 36% of the highly sedentary people were inactive. Therefore, the next step in research is to investigate whether movement behaviors cluster in patterns. The emergence of movement behavior patterns will provide insight into individuals' accumulation and distribution of movement behaviors during the day. Tremblay et al. described four hypothetical movement behavior patterns based on the distribution of movement behavior: 1. active and not sedentary; 2. active and sedentary; 3. inactive and not sedentary; and 4. inactive and sedentary [7]. Whether these patterns apply to the stroke population is currently unknown. Using these movement behavior patterns, individuals with an unfavorable patterns of behavior can be identified. Additionally, it will be important to investigate characteristics that help to differentiate among individuals with a favorable and unfavorable movement behavior pattern. This deeper understanding of the clustering patterns could support the development of personalized interventions to improve movement behavior during waking hours [69].

Although we expected to observe more changes in movement behavior outcomes based on the efforts of health care professionals, the willingness to change because of having experienced a stroke and the fact that recovery was feasible at the time of the study, only small changes in movement behavior outcomes occurred. In this sample, 46% of the population received physiotherapy care. In general, physiotherapy care focuses on regaining physical function and improving physical fitness [70]. However, improvements capabilities due to functional recovery will automatically improve ADLs, but will not automatically improve daily physical activity [71] or reduce SB. Additionally, as a general practice, all people with stroke in the Netherlands are included in primary care programs in general practice. However, in these programs there is limited attention for secondary prevention after stroke, especially physical activity [72,73]. Additionally, changing movement behavior is a complex process and cannot be triggered by merely providing information [74,75]. Therefore, specific interventions are needed to improve daily physical activity and decrease sedentary time. Particularly since the majority of our sample was sedentary and a substantial part was inactive, improving movement behavior is important and needs to be targeted to counterbalance increased cardiovascular risks. Additionally, the participants in this study had relatively minor stroke symptoms but nonetheless were highly sedentary, proportion of the sample was inactive. Although it is possible to modify daily physical activity and SB, it is not possible at present to suggest the superiority of a particular intervention approach [76].

A limitation of our study was that data regarding movement behavior during the day was obtained within three weeks after discharge. Therefore, it remains unknown whether

movement behavioral changes occur within the period immediately after discharge and three weeks later. Additionally, prestroke movement behavior during waking hours was not obtained. Therefore, it remains unknown whether people in this sample changed their movement behavior according to the behavior in the prestroke period. Another limitation was that sleep time during the day was not determined, and therefore, SB may have been overestimated. Last, our study included only participants who were directly discharged to the home-setting. Since, the majority of this population had minor stroke symptoms the results are not generalizable to a more severe stroke population that received inpatient rehabilitation first. However, our findings emphasize the importance of movement behavior changes since our sample had less severe stroke symptoms but still presented high levels of SB and low levels of MVPA.

Overall, the majority of people with stroke are highly sedentary, and a substantial proportion of this population is inactive in the first two months after discharge from hospital care based on continuous objective measurement for five weeks. Furthermore, their movement behavior remains fairly stable in this period. Based on movement behavior outcomes, distinctive subgroup trajectories were found. Although the people in this study had minor stroke symptoms, they were nonetheless highly sedentary, and a substantial portion was inactive. Therefore, changes movement behavior after discharge from the hospital are of paramount interest. Instead of providing information about changing movement behavior, personalized coaching interventions are needed. However, before such interventions take place, insight is needed into whether movement behavior during waking hours may cluster in patterns and which characteristics are related to an unfavorable movement behavior pattern in stroke survivors.

## Supporting information

**S1 File.** (Table A) Linear slopes and quadrics slopes with outcome values. (Table B) Distribution of individuals to different subgroups per movement behavior outcome expressed in percentages.
(DOCX)

## Acknowledgments

We would like to thank all participants for their contribution to the RISE-study. Furthermore, we would like to thank the staff of Catharina Hospital (Eindhoven), Jeroen Bosch Ziekenhuis ('s Hertogenbosch), Maxima Medisch Centrum (Veldhoven) and Sint-Jans Gasthuis (Weert) and we would like to thank Thirsa Koebrugge and Joeri Polman who helped with the data collection.

## Author Contributions

**Conceptualization:** Roderick Wondergem, Martijn F. Pisters, Martijn W. Heijmans, Eveline J. M. Wouters, Rob A. de Bie, Cindy Veenhof, Johanna M. A. Visser-Meily.

**Data curation:** Roderick Wondergem, Martijn F. Pisters.

**Formal analysis:** Roderick Wondergem, Martijn F. Pisters, Martijn W. Heijmans.

**Funding acquisition:** Roderick Wondergem, Martijn F. Pisters, Eveline J. M. Wouters, Cindy Veenhof, Johanna M. A. Visser-Meily.

**Investigation:** Roderick Wondergem, Martijn F. Pisters.

**Methodology:** Roderick Wondergem, Martijn F. Pisters, Martijn W. Heijmans, Rob A. de Bie, Cindy Veenhof.

**Project administration:** Roderick Wondergem.

**Software:** Martijn W. Heijmans.

**Supervision:** Martijn F. Pisters, Eveline J. M. Wouters, Rob A. de Bie, Cindy Veenhof, Johanna M. A. Visser-Meily.

**Validation:** Martijn W. Heijmans, Cindy Veenhof.

**Visualization:** Roderick Wondergem.

**Writing – original draft:** Roderick Wondergem, Martijn F. Pisters.

**Writing – review & editing:** Roderick Wondergem, Martijn F. Pisters, Martijn W. Heijmans, Eveline J. M. Wouters, Rob A. de Bie, Cindy Veenhof, Johanna M. A. Visser-Meily.

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
