## [Decision Letter · Decision Letter 0]

1 Oct 2019

PONE-D-19-23463

Movement behavior remains stable in stroke survivors within the first two months after returning home

PLOS ONE

Dear Mr. wondergem,

Thank you for submitting your manuscript to PLOS ONE. After careful consideration, we feel that it has merit but does not fully meet PLOS ONE’s publication criteria as it currently stands. Therefore, we invite you to submit a revised version of the manuscript that addresses the points raised during the review process.

Apart from the comments made by the reviewers the discussion is lacking a comparison with "healthy" individuals. You state that "the majority of the stroke survivors are inactive and highly sedentary" while report that they spend 0.7 h daily on at least moderate intensity which is approximately equivalent to 42 minutes of MVPA daily. This is around twice the amount compared to a general population. Thus the conclusion seems not supported by your data. Also you should focus on the validity issues raised by the reviewers.

We would appreciate receiving your revised manuscript by Nov 15 2019 11:59PM. To enhance the reproducibility of your results, we recommend that if applicable you deposit your laboratory protocols in protocols.io, where a protocol can be assigned its own identifier (DOI) such that it can be cited independently in the future. For instructions see: http://journals.plos.org/plosone/s/submission-guidelines#loc-laboratory-protocols

We look forward to receiving your revised manuscript.

Kind regards,

Patrick Bergman

Academic Editor

PLOS ONE

**Journal Requirements:**

2. Thank you for stating that “The funders had no role in study design, data collection and analysis, decision to publish, or preparation of the manuscript” in your financial disclosure.

Please also provide the name of the funders of this study (as well as grant numbers if available) in your financial disclosure statement.

**Comments to the Author**

1. Is the manuscript technically sound, and do the data support the conclusions?

Reviewer #1: Yes

Reviewer #2: Partly

2. Has the statistical analysis been performed appropriately and rigorously? 

Reviewer #1: Yes

Reviewer #2: Yes

3. Have the authors made all data underlying the findings in their manuscript fully available?

Reviewer #1: Yes

Reviewer #2: Yes

4. Is the manuscript presented in an intelligible fashion and written in standard English?

Reviewer #1: Yes

Reviewer #2: Yes

5. Review Comments to the Author

Reviewer #1: The authors of this manuscript tackle a highly important topic, that is changes in physical activity in the sub-acute phase after stroke. While numerous studies have demonstrated low levels of physical activity in the chronic phase after stroke little is known about the trajectory of physical activity over time. For that reason, as well as the large sample size and prolong follow-up of physical activity using accelerometers, this paper would make an important contribution to this field. There are some concerns though that I would like the authors to consider.

Major comments

1. The term “movement behavior” is broad and vague. This paper has clearly investigated different intensity domains of physical activity – therefore the terminology should be changed from movement behavior to physical activity throughout the manuscript.

2. Please clarify the validity of the Activ8 accelerometer to detect sedentary, light and MVPA in people with stroke.

3. The second aim; detecting possible subgroup trajectories within each outcome (i.e. different intensities of physical activity) using latent class growth analyses. To me, this section is statically driven and I would like the authors to justify this choice. For example, by explaining how detection of sub-groups following certain trajectory in physical activity would inform secondary prevention strategies post-stroke. Personally, I believe it is important to identify personal and functioning predictors of a “positive” vs “negative” (i.e. sedentary) physical activity trajectory post-stroke – and data from this trial seem to be able to contribute to this. Describing and contrasting the characteristics of stroke survivors following different trajectories could inform clinicians and researchers to target those of increased risk of sedentary and worsen cardio-vascular health. Taken together, to increase the influence of this paper, I think the rational and implication of the findings of the second aim should be clarified.

4. Does the results for the physical activity outcomes (sedentary, light and MVPA) reveal unique trajectories or is the results demonstrating a single trajectory of sedentary behavior post-stroke (see comment 11 below). Please clarify.

5. As shown in table 1, functioning and stroke severity data was also collected in this study. Please justify why changes in these outcomes was not considered in relation to changes in physical activity.

Abstract

6. Results; I would suggest to present the percentage of total wear time for each intensity (sedentary, light and MVPA) as it provides an informative summary of the distribution between intensity categories.

7. Result (first sentence): It is unclear whether this sentence refer to the entire time period for data collection or a certain time period.

Introduction

8. The introduction is overall well written, easy to follow, and introducing the reader to the research topic. My only comment is in line with my previous comment regarding aim 2, how would identification of certain subgroup trajectories of physical activity (without identifying who these individuals are) inform secondary stroke prevention?

Method

Recruitment and inclusion criteria are clearly stated.

9. Performance based tests of balance and gait were undertaken in the participants home. Berg Balance Scale and 5MWT requires some space to be performed, was it possible to perform these tests in all subject’s home in a standardized manner?

10. In table 1, data on the provision of physiotherapy care is provided, but I cannot find any information of these variables in the method section. Please a description of these variables including the nature, timing and definitions of these services as they could differ between health care systems.

Statistical analyses are sound and clearly described.

Results

11. Results are overall clearly presented. As pointed out in my previous comment, describing the characteristics of the stroke survivors following different trajectories would be informative. Also, to what extent does the trajectories overlap between the different PA outcomes? For instance, 90 participants were defined as highly sedentary, 92 as non-movers according to light PA and 89 as inactive according to MVPA. The weighted median sedentary bout length also adds up to about 90 if combining the “prolongers” (n=14) and intermediate groups (n=74). Are these PA outcomes revealing a single trajectory of sedentary behavior post-stroke occurring in approximately 65% of this sample? Or do the different PA intensities reveal unique patterns? I am curious about the authors thoughts on this potential overlap and it would be very interesting if data supporting or rejecting such overlap could be presented.

12. Please add information on time since stroke when the assessment of physical activity started – such data would allow comparison with similar longitudinal studies.

13. Figure 1 is blurry – please improve the quality. In addition, I would suggest adding 95% confidence intervals to the estimates.

Discussion

Relevant and clear – linked and contrasted to previous body of research.

14. First sentence (first paragraph): it is stated that the assessment took place “in the first 2 months after discharge” indicating a period of data collection of 10 weeks. Please rephrase.

15. The authors might need to revise the reasoning in the discussion regarding the unique trajectories for different intensity outcomes; e.g. “distinctive subgroup trajectories were found for each movement behavior outcome”, depending on whether they are unique patterns or not.

List of abbreviations

16. As a reader, it is rather demanding to keep track on all the short-forms used in this manuscript (n=16). I would recommend to only use short-forms of the most frequently used terms.

Reviewer #2: This is an interesting study with a relatively large n that examines the time course of activity levels during the first 2 months post stroke

Major:

Why was activity measured only during the time frame from 3 weeks to 2 months post stroke? Recovery post stroke is still likely happening past this time point. Also, were the participants still receiving home or out patient rehabilitation during this time (I see a brief amount of info on this in table 1, but there is no mention of it anywhere else, please elaborate on this, also was there any difference in activity levels based on receiving or not receiving rehab?). Both of these factors could greatly influence activity levels (time in sitting/lying, etc)? Related to this please provide information on exactly when post stroke the participants started wearing the devices and stopped. Just a blanket statement of 3 weeks post discharge from the hospital is not sufficient. This is important information in order to interpret the activity levels.

What are the clinical characteristics of the different subgroups (i.e. GS, BBS, etc.), are there any differences in these between the groups?

This was a very high functioning group in regards to mobility as evidenced by the mean gait speed of 1.03 m/s, this should be at least discussed as a limitation.

The Activ8 device has not been validated to identify energy expenditure in people with stroke, the article cited demonstrates its ability to identify postures (lying/sitting, standing, and walking. The authors use the outcomes time in SB, LP, and MVPA with associated METs. This is misleading as the device has not been validated to do this in people with stroke. Additionally, the method used by the authors to secure the device is not consistent with the study cited. To be more accurate the authors should use time spent in the different postures as their outcomes, not in SB, LP, MVPA.

The authors state that the small decrease in SB and increase in LPA are likely not clinically important. Although they are likely correct in this assessment they should provide some evidence of this.

Minor:

The statement associated with reference 4 in the first paragraph is misleading. Only systolic BP was found to impacted by sedentary lifestyle intervention; cardiovascular event rate mortality, diastolic blood pressure, or total cholesterol were not. Please revise.

The statement related to reference 14 in the second paragraph is also misleading. The article cited is a study protocol, they did not determine that interruption of sedentary bouts with activity reduces BP.

5th paragraph of the discussion, the point about the current approach is not clear and misleading. This was not an intervention study so there was no “approach” to lessen SB and increase activity levels. Also it is not clear how many participants were receiving rehabilitation services.

6. PLOS authors have the option to publish the peer review history of their article (what does this mean?). If published, this will include your full peer review and any attached files.

Reviewer #1: Yes: David Conradsson

Reviewer #2: No

---

## [Author Response · Author response to Decision Letter 0]

22 Nov 2019

We provide our response in the 'response to the reviewers'

---

## [Decision Letter · Decision Letter 1]

24 Dec 2019

PONE-D-19-23463R1

Movement behavior during waking hours remains stable in stroke survivors within the first two months after returning home

PLOS ONE

Dear Mr. wondergem,

Thank you for submitting your manuscript to PLOS ONE. After careful consideration, we feel that it has merit but does not fully meet PLOS ONE’s publication criteria as it currently stands. Therefore, we invite you to submit a revised version of the manuscript that addresses the points raised during the review process.

As reviewer three points out there are still a few things that you need to adress, most importantly

* The level of physical activity in these patients is high, even compared with healthy adults. The level of sedentary behaviour is also on the same level as healthy adults. Also the proportion of physical activity and sedentary behaviour is similar to that found among healthy adults and children (MVPA ~ 5% and SB ~ 70% of the measured time). The conclusions should be amended accordingly.

* The potential explanation to this may be thet the device used to measure physical activity is not validated in the population, thus there may be missclassifications present. This, however, do not change the conlusions regarding the change in physical activity and sedentary behaviour over time. This should be discussed

We would appreciate receiving your revised manuscript by Feb 07 2020 11:59PM. To enhance the reproducibility of your results, we recommend that if applicable you deposit your laboratory protocols in protocols.io, where a protocol can be assigned its own identifier (DOI) such that it can be cited independently in the future. For instructions see: http://journals.plos.org/plosone/s/submission-guidelines#loc-laboratory-protocols

We look forward to receiving your revised manuscript.

Kind regards,

Patrick Bergman

Academic Editor

PLOS ONE

Reviewers' comments:

Reviewer's Responses to Questions

**Comments to the Author**

1. If the authors have adequately addressed your comments raised in a previous round of review and you feel that this manuscript is now acceptable for publication, you may indicate that here to bypass the “Comments to the Author” section, enter your conflict of interest statement in the “Confidential to Editor” section, and submit your "Accept" recommendation.

Reviewer #1: All comments have been addressed

Reviewer #3: All comments have been addressed

2. Is the manuscript technically sound, and do the data support the conclusions?

Reviewer #1: Yes

Reviewer #3: Partly

3. Has the statistical analysis been performed appropriately and rigorously? 

Reviewer #1: Yes

Reviewer #3: I Don't Know

4. Have the authors made all data underlying the findings in their manuscript fully available?

Reviewer #1: Yes

Reviewer #3: No

5. Is the manuscript presented in an intelligible fashion and written in standard English?

Reviewer #1: Yes

Reviewer #3: Yes

6. Review Comments to the Author

Reviewer #1: The authors have done an excellent job responding to the issues raised and clarifying these in the manuscript. Except for a minor issue with regards to Table 4, I have no further comments or issues that I want to the authors to address.

Table 4 reports the result for the multivariate logistic regression -I would suggest to include the results for the univariate logistic regression as a separate table as well. Also, all independent variables need to be understandable from reading only the table itself (without looping back to the method section) – please clarify the cut-points used for each nominal variables (i.e. similar to Table 1).

Reviewer #3: Review

The line numbers are missing to accurately notify the remarks

The topic of this work is major in the management of stroke patients in the field of physical activity. Evaluating changes in physical activity levels and sedentary behaviour in the home is central to identifying the determinants of these changes in order to propose effective actions. The authors do not elaborate much on this in their introduction and that is a pity.

It is worth noting the authors' effort to have evaluated the participants in such a complete manner and to have carried out such a consistent follow-up on such a large group of patients.

The major limitation of this work is the lack of validation of activ8 for the estimation of energy expenditure or METs in stroke subjects. This is a major limitation because the entire classification process of the study concerns the quality of the device to differentiate the different categories of physical activity intensity performed by the individual stroke at home. Despite a quality work performed by Fanchamps et al showing a validity of physical activity detection, the transposition of METs on activities identified by Activ8 from healthy subject populations is inadequate cf Serra et al 2016, Compagnat et al 2018. In addition, this device uses a uniaxial accelerometer with an algorithm developed on a healthy subject. However, we now know that estimates of this type of device are not reliable in stroke subjects for the same reasons explained below. It therefore seems necessary to explore the validity of a device that has not been validated before being used in clinical research.

If the reader accepts this uncertainty I have developed some remarks about the manuscript.

Abstract results: I would add if possible which subgroups are identified since this is part of the objectives of the study.

Introduction

Your assumptions are contrary to what is reported in the literature, particularly in the literature review of English et a et Rand et al l, which clearly shows that people with stroke tend to be more sedentary. In addition, we observe in the literature and in our clinical experience a strong association between physical activity level and functional abilities, mood disorders, self-efficacy that you do not address at all in your introduction..

Methodology

Design and participant

How can you justify including only stroke survivors with such a high level of autonomy? This is excessively regressive and unrepresentative of strokes treated in rehabilitation unit.

Measurements and procedures :

Specify that the evaluation procedures follow this paragraph, otherwise the reader may be frustrated.

Accelerometer

The absence of validation of activ8 in the stroke subject in terms of energy expenditure is a major limitation for the interpretation of the activity intensities reported by the device. Indeed, these intensities are based on the energy expenditure for the activity which is strongly different between healthy individuals and undivided AVC.

Results :

What was the number of refusals to participate in the study? Wouldn't the patients who agreed to participate in the study be those who consider themselves the least sedentary or inactive? A flow chart seems necessary to inform the reader about these elements.

What was the method used to define the different sedentary classes (low/high)? Is this related to the statistical analysis you describe in the method section? What is the clinical relevance between these 2 separate categories of 2 hours of low-intensity activities when both are sedentary by definition. Same for the activity categories (high, moderate, inactive)...

The authors provide part of the answer in the results paragraph but this is not the place for that. It would be necessary to place this information back in the method section. Moreover, to my knowledge, I do not believe that the clinical relevance of this type of category has been demonstrated in individuals with stroke.

The last paragraph of the results are, in my opinion, extremely interesting and need to be developed because there is a strong stake in knowing what are the associations between sedentary behaviours, autonomy and social participation scores and mood disorders. The assessment of possible changes in sedentary behaviours and physical activity levels associated with these different dimensions that you have assessed (impairments, activity, participation) as well as those that may or may not benefit from physiotherapy would be all the more original.

To be honest, I expected the authors to establish the subgroups on the clinical characteristics of the participants and not on this type of statistically constructed categories. This limits its clinical interest.

Discussion

Is the high value of MVPA in these individuals not related to the particular profile of patients you have included: undivided with very little neuronal sequelae?

Table 1 :

I don't think we can report the SIS and BBS values on average and SD since they are not continuous variables.

7. PLOS authors have the option to publish the peer review history of their article (what does this mean?). If published, this will include your full peer review and any attached files.

Reviewer #1: Yes: David Moulaee Conradsson

Reviewer #3: No

---

## [Author Response · Author response to Decision Letter 1]

5 Feb 2020

We provided the response in the attached file

---

## [Editor Report · Decision Letter 2]

11 Feb 2020

Movement behavior remains stable in stroke survivors within the first two months after returning home

PONE-D-19-23463R2

Dear Dr. wondergem,

We are pleased to inform you that your manuscript has been judged scientifically suitable for publication and will be formally accepted for publication once it complies with all outstanding technical requirements.

With kind regards,

Patrick Bergman

Academic Editor

PLOS ONE
---

## [Editor Report · Acceptance letter]

13 Mar 2020

PONE-D-19-23463R2 

Movement behavior remains stable in stroke survivors within the first two months after returning home 

Dear Dr. Wondergem:

I am pleased to inform you that your manuscript has been deemed suitable for publication in PLOS ONE. Congratulations! Your manuscript is now with our production department. 

With kind regards,

on behalf of

Dr. Patrick Bergman 

Academic Editor

PLOS ONE